# Combination of Secondary Plant Metabolites and Micronutrients Improves Mitochondrial Function in a Cell Model of Early Alzheimer’s Disease

**DOI:** 10.3390/ijms241210029

**Published:** 2023-06-12

**Authors:** Lukas Babylon, Julia Meißner, Gunter P. Eckert

**Affiliations:** Laboratory for Nutrition in Prevention and Therapy, Biomedical Research Center Seltersberg (BFS), Institute of Nutritional Sciences, Justus-Liebig-University, Schubertstr. 81, 35392 Giessen, Germany; lukas.babylon@ernaehrung.uni-giessen.de (L.B.); julia.meissner@ernaehrung.uni-giessen.de (J.M.)

**Keywords:** hesperetin, magnesium-orotate, folic acid, caffeine, kahweol, cafestol, ROS, Alzheimer’s disease, mitochondrial dysfunction, amyloid-beta

## Abstract

Alzheimer’s disease (AD) is characterized by excessive formation of beta-amyloid peptides (Aβ), mitochondrial dysfunction, enhanced production of reactive oxygen species (ROS), and altered glycolysis. Since the disease is currently not curable, preventive and supportive approaches are in the focus of science. Based on studies of promising single substances, the present study used a mixture (cocktail, SC) of compounds consisting of hesperetin (HstP), magnesium-orotate (MgOr), and folic acid (Fol), as well as the combination (KCC) of caffeine (Cof), kahweol (KW) and cafestol (CF). For all compounds, we showed positive results in SH-SY5Y-APP_695_ cells—a model of early AD. Thus, SH-SY5Y-APP_695_ cells were incubated with SC and the activity of the mitochondrial respiration chain complexes were measured, as well as levels of ATP, Aβ, ROS, lactate and pyruvate. Incubation of SH-SY5Y-APP_695_ cells with SC significantly increased the endogenous respiration of mitochondria and ATP levels, while Aβ_1–40_ levels were significantly decreased. Incubation with SC showed no significant effects on oxidative stress and glycolysis. In summary, this combination of compounds with proven effects on mitochondrial parameters has the potential to improve mitochondrial dysfunction in a cellular model of AD.

## 1. Introduction

Alzheimer’s disease (AD) has now been placed on the WHO’s list of global health priorities. Since its first description in 1906, there is still no effective treatment for the disease [1]. In 2018, 1.73% of the population in the European Union was living with dementia and this figure is expected to reach 2.0% by 2050 [2]. AD is the most common form of dementia and is on the rise due to an aging population [3]. AD is characterized by a progressive loss of memory, speech, personality and behavior, which is accompanied by limitations in the quality of life [3]. Signs including the presence of amyloid beta (Aβ), which accumulates extracellularly, leading to plaque formation and hyperphosphorylation of tau protein characterize AD [4,5]. Furthermore, AD leads to impaired glucose metabolism [6] and increased reactive oxygen species (ROS) production, which in turn leads to increased oxidative stress [7,8]. Mitochondrial dysfunction (MD) was identified as playing a central role in the etiology of AD, leading to reduced metabolism, increased ROS, lipid peroxidation and finally to apoptosis [9,10]. The first sign of MD is a decrease of enzymes from the oxidative metabolism [11,12], leading to a limited function of the electron transport chain (ETC) and thus to a reduced complex activity of complexes I and IV. This in turn leads to a lower membrane potential and to lower ATP levels [13,14]. These defects lead to increased ROS production and defects within the mitochondrion, which contributes to an increase in mitochondrial dysfunction [15]. 

One way to support the various defects in AD is through the use of secondary plant compounds, especially flavonoids [16,17,18]. The flavonoid hesperetin (HstP) is an aglycon of hesperedin and is mainly found in the peel of oranges and other citrus fruit [19]. The bioavailability of HstP is limited; only about 20% of it is absorbed. On the other hand Hesperedin, the glycosylated form, is almost completely absorbed [20,21]. However, its possible ability to cross the blood–brain barrier makes it a potential candidate for neurodegenerative diseases [22]. HstP was shown to have a protective effect on neurons previously damaged by Aβ [23]. Hesperetin improve cognitive abilities by increasing BDNF levels in rats. Furthermore, it increases the activities of CAT, SOD, GRX and GPX, and cytoprotective effects of hesperetin against Aβ-induced impairment of glucose transport have been reported [24]. We recently have shown that HstP reduces ROS levels in SH-SY5Y-APP_695_ cells without affecting mitochondrial function or Aβ_1–40_ level [25]. In another study, we showed that magnesium-orotate (MgOr) and folic acid (Fol) reduced Aβ, soluble alpha APP (sAPPα) and lactate levels in SH-SY5Y-APP_695_ cells [26].

Folate is an important product in metabolism, and when deficient, leads to insufficient DNA and mtDNA synthesis and stability, which is why it can lead to oxidative stress, a process associated with Alzheimer’s disease. This leads to the worsening of the neuronal state and increased cell death. Deficiency leads to impaired methylation of enzymes and promoter regions of genes involved in AD. [25,26]. MgOr is the magnesium salt of orotic acid, showing enhanced bioavailability. Orotic acid is a key substance in biosynthetic pathways, as well as in the synthesis of glycogen and ATP [27]. Mg is involved in many synthesis processes, as well as in maintaining physiological nerve and muscle function [28,29,30]. Many different mechanisms related to Mg and AD are discussed. It is thought to positively influence various inflammatory pathways and to influence APP processing towards α-secretase, as well as preventing Aβ from crossing the blood–brain barrier [29,31]. Another product discussed in connection with AD are ingredients of coffee, particular caffeine, kahweol and cafestol. Coffee is one of the most commonly consumed beverages [32]. Coffee is supposed to have cardio-, hepato- and neuroprotective properties [33]. The consumption of coffee and caffeine is associated with a lower risk of AD [34,35,36]. After oral intake, caffeine is 99% absorbed and peaks in the bloodstream after 30–45 min [37]. Cafestol is absorbed at about 70%, while kahweol has a slightly higher bioavailability [38]. All three substances show the ability to cross the blood–brain barrier [39]. Data on kahweol and cafestol are scarce, but there is evidence that both are neuroprotective [40,41,42]. Our preliminary data show that a combination of caffeine, kahwol and cafestol (KCC) affects glycolysis in SH-SY5Y-APP_695_ cells (unpublished data). 

Based on the previous experiments on various dietary compounds in SH-SY5Y-APP695 cells, which showed different beneficial effects, the hypothesis was tested whether a combination of these compounds would also have beneficial activity and possibly show synergistic effects. HstP improves mitochondrial dysfunction [43], MgOr and Fol reduce Aβ levels within cells [26] and caffeine, kahwol and cafestol (KCC) affects glycolysis (unpublished data). A combination of all compounds (SC) in meaningful doses was tested in the SH-SY5Y-APP_695_ cell model regarding mitochondrial function, oxidative stress, Aβ production and glycolysis.

## 2. Results

### 2.1. Characterization of the Cellular Model

SH-SY5Y-APP_695_ cells are an established model for early phase AD [44,45]. SH-SY5Y-APP_695_ cells show significantly decreased ATP levels and respiration compared to SH-SY5Y-MOCK cells (Figure 1A,B). Furthermore, in SH-SY5Y-APP_695_ cells, Aβ levels are significantly increased (Figure 1C) and ROS production is significantly increased (Figure 1D) compared to control cells.

### 2.2. Overview

The SC used in this study is a combination of different compounds, which have been previously tested as single substances or combinations in SH-SY5Y-APP_695_ cells. The Table 1 compares the effects of the individual substances and their concentrations with the results of the SC reported herein.

### 2.3. Mitochondrial Parameter

First, we examined the cells for various mitochondrial functions, including ATP levels, respirometry and related citrate synthase activity. Incubation with SC resulted in a significant increase in ATP levels compared to control (*p* = 0.0031, Figure 2A). Since the increased ATP levels may be due to a change in the activity of respiration chain complexes, respiration was investigated next. A significant increase in endogenous respiration of SC compared to SCTR was found (*p* = 0.0314). Except for complex II and leak II, the O_2_ consumption of SC tended to be higher than that of SCTR (Figure 2B). However, the changes were not significantly different. Moreover, there was no difference in the mitochondrial mass marker citrate synthase activity between the groups (Figure 2C).

### 2.4. Aβ_1–40_ Level

To investigate the Aβ levels, the cells were incubated with SC or SCTR for 24 h (Figure 3). SC significantly decreased the Aβ levels of the cells compared to the control (*p* = 0.0217).

### 2.5. ROS

To investigate the effects of incubation on ROS levels, cells were incubated with the respective substance for 24 h (Figure 4). No significant differences were found.

### 2.6. Glycolysis

To investigate the effects of the incubations on glycolysis metabolism, the lactate and pyruvate values and their ratio were determined (Figure 5). There were no significant differences, although the lactate values of the SC tended to be higher than those of the control (*p* = 0.0983). In contrast, the pyruvate values of the SC tended to be lower than those of the control. This results in a lower ratio for the control (*p* = 0.0509).

## 3. Discussion

In the following manuscript, we investigate the effects of HstP, MgOr, Fol, Cof, KW and CF as a cocktail in a human cell model of early AD. All of the previously listed compounds have shown effects in the area of mitochondrial dysfunction, Aβ levels and glycolysis in previous studies we conducted. Our question in this regard was whether all the substances, combined in a cocktail, address all the areas mentioned. The easy availability of the substances offers a good opportunity to intervene in early AD symptoms.

Mitochondrial dysfunction is one of the core issues in the development of AD, as described earlier, and is characterized by a decrease in ATP levels [13,14,46]. HstP elevated ATP levels in SH-SY5Y-APP_695_ cells in our former studies [43]. Similar findings were reported in differentiated human myotubes by Biesemann et al. [47]. Mg and Fol are also involved in the provision of ATP. Deficiencies of these nutrients leads to a decrease in ATP production [29,48]. Deficits are also evident in AD, in which Mg and Fol levels are found to be reduced in brain tissues of patients [49,50,51,52]. In our studies using SH-SY5Y-APP_695_ cells as an AD model, no increase in ATP levels was observed from the single administration of MgOr and Fol [26] The combination of the three coffee ingredients (Cof, KW, and CF) increased ATP levels [25], but Cof as single compound had no effect on ATP levels as shown in our experiments. However, it has been described in the literature that Cof affects ATP levels in HepG2 cells and smooth muscle cells [53,54]. 

The OXPHOS in mitochondria is built up by the activity of respiration chain complexes and is the main cellular source of ATP production [55,56]. The administration of SC significantly increased endogenous respiration in mitochondria. Furthermore, the activity of the individual complexes of the respiration chain was increased compared to the control. Increased respiration could be explained by the individual substances in the mixture (SC) or by the interaction of them. Activation of the Nrf-2 pathway by HstP represents one conceivable option [57,58]. Reduction of Nrf-2 results in decreased MMP, decreased ATP levels and impaired mitochondrial respiration [59]. A reduction of Aβ levels by MgOr and Fol could also be a possible explanation [26]. This could reduce the impairment of complexes by Aβ. In our former experiments, KCC had no effect on OXPHOS, so the described effect for SC could be due to the other substances in the mixture.

Next, we investigated whether the combination of all tested compounds had any effect on ROS levels in SH-SY5Y cells (Figure 4). It was found that the ROS levels of the cocktail did not decrease. HstP had no ROS-reducing effect, as shown in our other study [43]. It is possible that the amount of the substances means that no ROS-reducing effect occurred. Fol and MgOr had no effect on ROS levels (Figure 3), whereas the literature suggests a ROS reducing effect from Mg [60], Fol [61], Cof [62], KW [63] and CF [64]. However, the described effects are difficult to compare, either because they are deficit models or because models other than our SH-SY5Y cell model were used. 

Next, we took a closer look at Aβ levels (Figure 3). We found that SC significantly lowered Aβ_1–40_ compared to the control. Considering the SC results, there is an additive effect on Aβ_1–40_ levels. Of the compounds we tested, only MgOr and Fol had the potential to significantly lower Aβ levels [26], but in combination as SC, all substances lowered Aβ levels. Indeed, all our tested substances show the same potential in the literature [57,65,66,67]. The question now is whether the effect we observed is due to the combination of MgOr and Fol alone, or whether all substances are exerting their potential. It is possible that the combined effect, similar to the effect of the individual substances [66,67], results in suppression of BACE1 and APP expression and a greater reduction in β-secretase activity than we observed with MgOr and Fol [26]. This should be confirmed in further experiments.

As a final point, SC was examined for the effects on glycolysis (Figure 5). Although data did not reach significance, the incubation tended to increase lactate levels, with a concomitant decrease in pyruvate levels and an L/P in favor of lactate. Based on this observation, it can be assumed that the cells tend to increased anaerobic respiration due to the SC. It should be noted that MgOr and Fol, reduced Aβ levels, but did not affect glycolysis in our former experiments [26]. It has been shown that cells protect themselves from Aβ by increasing anaerobic respiration, initiating the Warburg effect [68,69,70]. This manifests itself in an increase in anaerobic respiration and an increase in lactate and a reduction in OXPHOS when the cells switch from aerobic to anaerobic respiration [71]. However, in our experiments OXPHOS increased.

In summary, SC has the potential to have a positive effect on AD, especially the increased ATP levels in combination with the lowered Aβ levels. Since SC consists of a variety of substances, it would be interesting for future studies to create combinations of two or three substances tested here and to investigate their effects on our parameters. 

## 4. Materials and Methods

### 4.1. Cell Culture

Human SH-SY5Y-APP_695_ cells were incubated at 37 °C in DMEM in an atmosphere of 5% CO_2_. DMEM was mixed with 10% heat inactivated fetal calf serum, streptomycin, penicillin, hygromycin, non-essential amino acids and sodium pyruvate. The cells were split on average every 3 days and used for experiments when they reached 70–80% growth.

### 4.2. Cell Treatment

Cells were incubated with a cocktail for 24 h each unless indicated. This cocktail contains a mixture of hesperetin 10 µM (HstP), magnesium orotate 200 µM (MgOr), folic acid 10 µM (Fol), caffeine 50 µM (Cof), cafestol 1 µM (CF) and kahweol 1 µM (KW) (together SC). The control (SCTR) was a mixture of solvents in which the substances were dissolved ethanol, DMSO and NaOH. 

### 4.3. ATP Assay

To determine ATP levels, an ATP luciferase assay was performed. Light is generated by ATP and luciferin in the presence of luciferase. The test was performed using the ATPlite Luminescence Assay System (PerkinElmer, Rodgau, Germany).

### 4.4. Cellular Respiration

Respiration in SH-SY5Y_695_ cells was measured using an Oxygraph-2k (Oroboros, Innsbruck, Austria) and DatLab 7.0.0.2. Cells were treated according to a complex protocol developed by Dr. Erich Gnaiger [72]. They were incubated with different substrates, inhibitors and uncouplers. First, cells were washed with PBS (containing potassium chloride 26.6 mM, potassium phosphate monobasic 14.705 mM, sodium chloride 1379.31 mM and sodium phosphate dibasic 80.59 mM), then scraped into mitochondrial respiratory medium (MiRO5) developed by Oroboros [72]. Cells were then centrifuged, resuspended in MiRO5 and diluted to 106 cells/mL. After 2 mL of cell suspension was added to each chamber and endogenous respiration was stabilized, cells were treated with digitonin (10 µg/106 cells) to permeabilize the membrane, leaving the outer and inner mitochondrial membranes intact. OXPHOS was measured by addition of complex I and II substrates malate (2 mM), glutamate (10 mM) and ADP (2 mM), followed by succinate (10 mM). The stepwise addition of carbonyl cyanide-4-, before being evaluated by measuring R123 fluorescence (trifluoromethoxy) phenylhydrazone, showed the maximum capacity of the electron transfer system. Rotenone (0.1 mM), a complex I inhibitor, was injected to measure complex II activity. Oligomycin (2 µg/mL) was then added to measure leak respiration. Inhibition of complex III via the addition of antimycin A (2.5 µM) was used to determine residual oxygen consumption, which was subtracted from all respiration states. The activity of complex IV was measured via the addition of N, N, N′-tetramethyl-p-phenylenediamine (0.5 mM) and ascorbate (2 mM). To measure the sodium autoxidation rate, azide (≥100 mM) was added. Subsequently, complex IV respiration was corrected for autoxidation. 

### 4.5. Citrate Synthase Activity

Cell samples from respirometry were frozen away at −80 °C for citrate synthase determination. Samples were spiked and mixed with the reaction mix (0.31 mM acetyl coenzyme A, 0.1 mM 5,5′-dithiol-bis-(2-nitrobenzoic acid) (DTNB), 50 μM EDTA, 0.5 mM oxaloacetate, 5 mM triethanolamine hydrochloride and 0.1 M Tris-HCl) and warmed to 30 °C for 5 min. Samples of 40 µL each were applied as a triplet, 110 µL of reaction mix was added and the absorbance was measured.

### 4.6. Protein Quantification

To determine the protein content of the samples, a PierceTM Protein Assay Kit (Thermo Fisher Scientific, Waltham, MA, USA) was used and the experiment was performed according to the manufacturer’s instructions, using bovine serum albumin as the standard.

### 4.7. Aβ_1–40_ Measurement

After 24 h incubation, Aβ_1–40_ was determined using an HTRF amyloid beta 1–40 kit (Cisbio, Codolet, France). The protocol has been described previously [43]. Aβ_1–40_ levels were normalized against protein levels.

### 4.8. ROS Quantification

A DCFDA/H2DCFDA kit was used to determine ROS levels (ab113851; Abcam, Cambridge, UK). Cells were seeded in 96-well plates and incubated for 24 h. The experiment was then performed according to the manufacturer’s instructions.

### 4.9. Lactate and Pyruvate Measurement

Frozen cell samples, which were previously incubated for 24 h, were thawed at room temperature. An assay kit (MAK071, Sigma Aldrich, Darmstadt, Germany) was used to determine pyruvate levels. For the determination of lactate values, a lactate assay kit was used (MAK064, Sigma Aldrich, Darmstadt, Germany). The values were normalized to the protein content.

### 4.10. Statistics

Unless otherwise stated, values are presented as mean ± standard error of the mean (SEM). Statistical significance was defined for *p* values * *p* < 0.05, ** *p* < 0.01, *** *p* < 0.001 and **** *p* < 0.0001. Statistical analyses were performed by applying student’s unpaired *t*-test and one-way ANOVA with Tukey´s multiple comparison post hoc test (Prism 9.5 GraphPad Software, San Diego, CA, USA). 

## 5. Conclusions

In our former experiments using SH-SY5Y-APP_695_ cells, we showed that HstP lowered the ROS level, MgOr and Fol lowered Aβ levels, and the combination of coffee ingredients affected glycolysis. In the current study, we addressed the question of whether a combination of these compounds have additive effects on the tested parameters. We found that the cocktail increased ATP levels and decreased Aβ levels. However, no advantage of the combination over the individual substances was found.

## Figures and Tables

**Figure 1 ijms-24-10029-f001:**
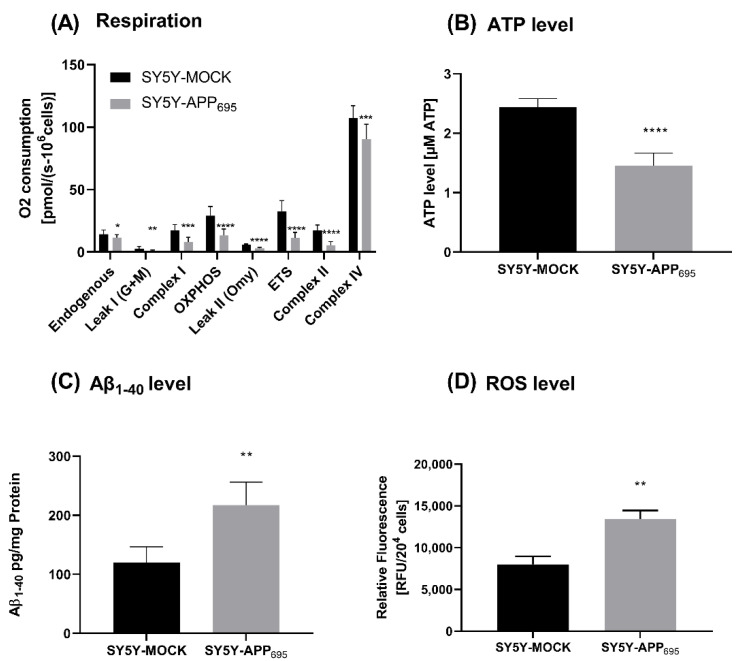
Respiration, ATP, Aβ_1-40_ level and ROS level of SH-SY5Y-APP695 cells compared to SH-SY5Y-MOCK cells. (**A**) Respiration of SH-SY5Y-MOCK and -APP cells adjusted to cell count. Respective mean values ± SD are shown. Significance was determined using Student’s unpaired *t*-test, N = 12. (**B**) SH-SY5Y-APP_695_ cells exhibit reduced ATP levels compared to SH-SY5Y-MOCk control cells. ATP levels were determined after 24 h seeding by bioluminescence assay. Respective mean values ± SD are shown. Significance was determined using Student’s unpaired *t*-test, N = 8. (**C**) Concentration of Aβ_1–40_ levels in SH-SY5Y-APP_695_ cells compared to SH-SY5Y-MOCK cells. Significance was determined using Student’s unpaired *t*-test. N = 6. (**D**) ROS levels in RUF/20^4^ cells in SH-SY5Y-APP_695_ cells compared to SH-SY5Y-MOCK cells. Respective mean values ± SD are shown. Significance was determined using Student´s unpaired *t*-test, N = 8. * *p* < 0.05, ** *p* < 0.01, *** *p* < 0.001, and **** *p* < 0.0001.

**Figure 2 ijms-24-10029-f002:**
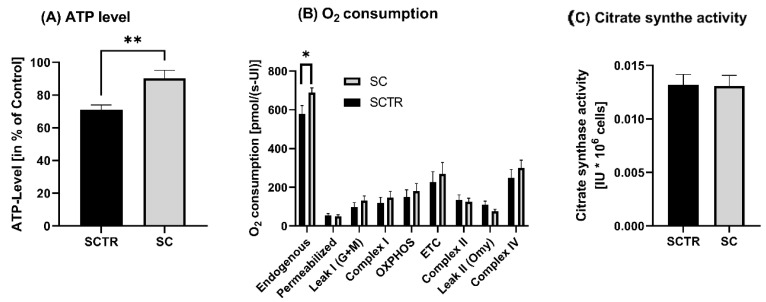
ATP level, respiration and citrate synthase activity of SH-SY5Y-APP_695_ cells incubated for 24 h with SC. (**A**) ATP level of SH-SY5Y-APP_695_ cells incubated with SC. Cells treated with cell culture medium served as control (100%). N = 11. (**B**) Respiration of SH-SY5Y-APP_695_ cells incubated with SC compared to the control. SH-SY5Y-APP_695_ cells adjusted to international units (IU) of citrate synthase activity. N = 16. (**C**) Citrate synthase activity of SH-SY5Y-APP_695_ cells incubated with SC compared to control. N = 16. Significance was determined using Student’s unpaired *t*-test. Data are displayed as the mean ± SEM. * *p* < 0.05, ** *p* < 0.01, SC = cocktail, SCTR = control.

**Figure 3 ijms-24-10029-f003:**
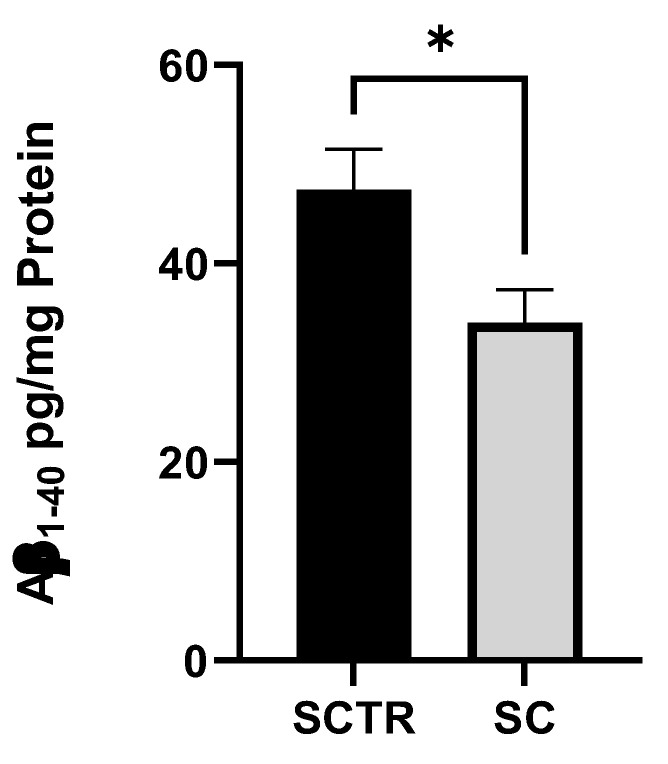
Effect of SC on the Aβ_1–40_ level of SH-SY5Y-APP_695_ cells incubated for 24 h. N = 9. Aβ_1-40_ levels were adjusted to the protein content. Significance was determined using Student’s unpaired *t*-test. Data are displayed as the mean ± SEM. * *p* < 0.05 SC = cocktail, SCTR = control.

**Figure 4 ijms-24-10029-f004:**
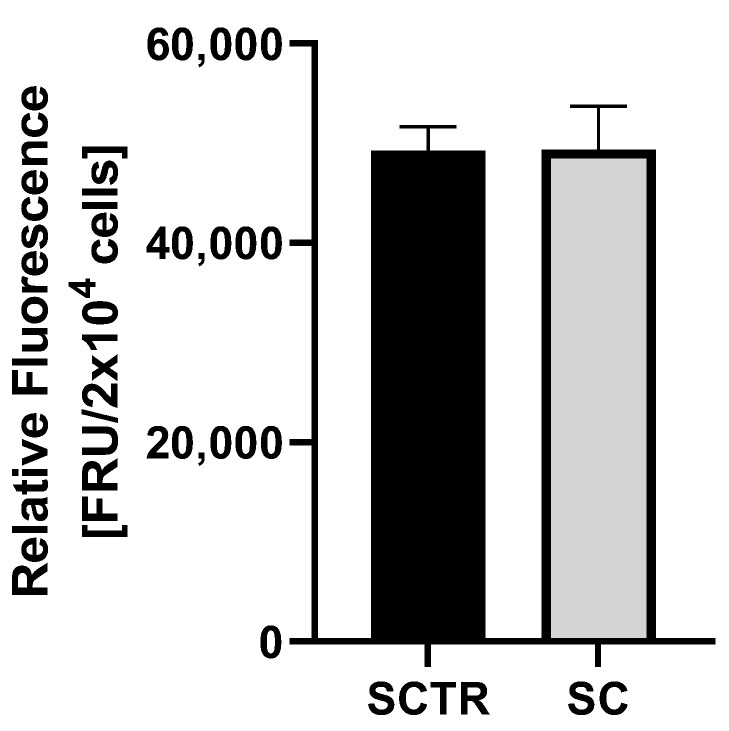
ROS measurement of SH-SY5Y-APP_695_ cells after incubation with SC for 24 h. N = 8. Significance was determined using Student’s unpaired *t*-test. Data are displayed as the mean ± SEM.

**Figure 5 ijms-24-10029-f005:**
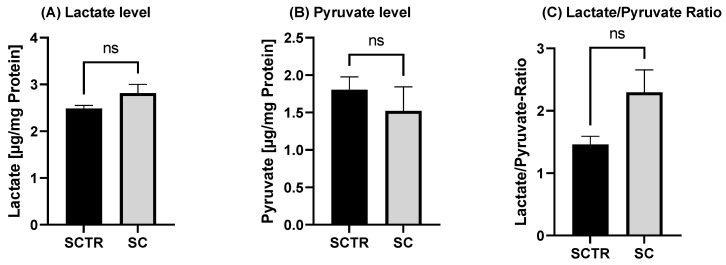
Effect of the on lactate and pyruvate level after 24 h of incubation with SC or the control. (**A**) Lactate level of SH-SY5Y-APP_695_ cells after the incubation compared to the control. (**B**) Pyruvate level of SH-SY5Y-APP_695_ cells after the incubation with SC compared to the control. (**C**) Lactate to pyruvate ratio. N = 10. Levels were adjusted to the protein content. Significance was determined using Student’s unpaired *t*-test. ns = not significant. Data are displayed as ±SEM.

**Table 1 ijms-24-10029-t001:** Effects of the combination of dietary compounds tested herein compared with the effects of the individual components in the SH-SY5Y-APP_695_ cells.

	HstP	MgOr + Fol	Cof + KW + CF	SC
ATP level	↑	=	↑	↑
OXPHOS	=	=	=	↑
ROS	↓	=	=	=
Aβ level	=	↓	=	↓
Lactate	=	↓	=	=
Pyruvate	=	=	↑	=
L/P Ratio	=	=	↓	=
Source	[43]	[26]	[25]	

↑ = significant increase; ↓ = significant decrease; = no significant change; L/P = lactate-pyruvate-ratio; Concentration of tested substances were HstP (10 µM), MgOr (200 µM), Fol 10 µM), Cof (50 µM). KW (1 µM), CF (1 µM) and SC (all concentration combined).

## Data Availability

The data presented in this study are available on request from the corresponding author.

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
