# Peer review of "Combination of Secondary Plant Metabolites and Micronutrients Improves Mitochondrial Function in a Cell Model of Early Alzheimer’s Disease"

_ijms, 2023, doi:10.3390/ijms241210029_

Round 1

Reviewer 1 Report

The topic of the manuscript  sounds quite interesting and innovative. However, significant failures cannot be overcome. I suggest to improve introduction and discussion, to clarify better your results and methods, to check your figures (some labels not legible and panels with different styles A, A), etc.) and, above all, some additional experiments are needed to enrich and strengthen the novelty of the manuscript.

Several mistakes on English language and editing are detectable throughout the text and figures. It is recommended an extensive revision.

Author Response

Reviewer 1:

The topic of the manuscript  sounds quite interesting and innovative. However, significant failures cannot be overcome. I suggest to improve introduction and discussion, to clarify better your results and methods, to check your figures (some labels not legible and panels with different styles A, A), etc.) and, above all, some additional experiments are needed to enrich and strengthen the novelty of the manuscript.

Several mistakes on English language and editing are detectable throughout the text and figures. It is recommended an extensive revision.

Thank you very much for your criticism. We have made some changes. We are aware that further experiments would always be helpful to increase the significance. With this study, we wanted to find out whether the experiments previously performed in our studies, achieved the same or similar results. Therefore, in this study, we limited ourselves only to the experiments that showed significant results in our previous studies.

Unfortunately, we could not confirm that our cocktail showed the same results in mitochondrial dysfunction (ROS), Aβ, and glycolysis as in our other studies.

For future studies, based on this study, we will limit ourselves to a smaller number of compounds to show possible additive effects.

Reviewer 2 Report

The current manuscript presented by Babylon et al. reported the combinational effect of several dietary compounds on early phase Alzheimer’s Disease cell model. The scope of this work fits the journal and the topic of the special issue very well. The authors have covered and summarized several compounds that could be beneficial in early Alzheimer’s Disease, which could be a useful reference for future research. Further experiments on the combinational effect are also inspiring, which exhibits the value to explore more details regarding their synergistic effects. The overall quality of this work as a quick and short report is good but I have a few comments would like to share, which are stated below:

1. Delivery efficiency of mentioned compounds in this study. Like many other compound studies in the fields of drug, clinic, etc., in vitro cell model has its own advantages and limitations. Even in the field of nutrition for application in prevention and therapy, delivery efficiency should also be an important topic when evaluating compound effects. The authors have discussed and summarized multiple dietary compounds in this study. I believe it will be helpful if the authors could supplement additional references in the introduction section to illustrate how possible these dietary compounds can be delivered to the brain after absorbance via the digestive system or other pathways, if possible. The reason I believe it is necessary is that their beneficial effects will possibly not happen in vivo at all if some of these compounds cannot be delivered to our nervous system in vivo.

2. Unpublished data mentioned in the article. If no specific reason, the authors are encouraged to show their unpublished data results in the supplementary materials. These results may also be helpful as a future reference.

3. Combinational effect in a smaller range such as two or three compounds. As the authors mentioned in their article, the combinational effect of two or three compounds may also be necessary to analyze and understand their synergetic effects precisely. The current combinational effect study has included too many compounds and the results shown by the authors cannot clearly show all potential synergetic effects among these compounds. Therefore, the authors cannot sufficiently explain the differences in the cocktail treatment compared to individual compounds. Here I just simply want to emphasize that this point may limit the impact of this article. Hopefully, in the future study, the authors could provide more details.

4. Wording issue. The term “SCTR” is not defined in the first place and I found its meaning in the caption for figure 2. If there is no specific reason, it would be better to use straightforward and common words such as treated/cocktail and control to describe your groups. Creating new abbreviations like SC and SCTR could be confusing and lower the readability of your research article. In addition, the authors' first explanation of SCTR is in the caption of their figures, which may not be obvious enough to readers. 

The overall quality of English is good but multiple typos and grammar errors have been detected. These typos and errors are listed below:

1. Introduction section, page 1 line 39: “… leading to to reduced…” has repeating words.

2. Introduction section, page 2 line 47: “The flavonoid is hesperetin (HstP) is an ……” has grammar error.

3. Introduction section, page 2 line 68: “… towards _-secretase, as well as preventing A_ from crossing ……” has two typos displaying special symbols.

4. Introduction section, page 2 line 82: “… oxidative stress, A_ production, ……” has a typo displaying a special symbol.

5. Figure 3, page 5 line 143: “… as the mean +/- SEM. *P>0.05 ……” has a typo. It should be p<0.05.

6. Discussion section, page 7 line 212-213: “SC was examined for it effects on ……. Although data were did not reach ……” has two grammar errors.

Author Response

Reviewer 2:

The current manuscript presented by Babylon et al. reported the combinational effect of several dietary compounds on early phase Alzheimer’s Disease cell model. The scope of this work fits the journal and the topic of the special issue very well. The authors have covered and summarized several compounds that could be beneficial in early Alzheimer’s Disease, which could be a useful reference for future research. Further experiments on the combinational effect are also inspiring, which exhibits the value to explore more details regarding their synergistic effects. The overall quality of this work as a quick and short report is good but I have a few comments would like to share, which are stated below:

  1. Delivery efficiency of mentioned compounds in this study. Like many other compound studies in the fields of drug, clinic, etc., in vitro cell model has its own advantages and limitations. Even in the field of nutrition for application in prevention and therapy, delivery efficiency should also be an important topic when evaluating compound effects. The authors have discussed and summarized multiple dietary compounds in this study. I believe it will be helpful if the authors could supplement additional references in the introduction section to illustrate how possible these dietary compounds can be delivered to the brain after absorbance via the digestive system or other pathways, if possible. The reason I believe it is necessary is that their beneficial effects will possibly not happen in vivo at all if some of these compounds cannot be delivered to our nervous system in vivo.

Thank you for the comments. We have added the following sources to address bioavailability and to show that our compounds show the ability to overcome the blood-brain barriers. Thus, we hope that a possible transfer from cell models to humans would be possible.

“The bioavailability of HstP is limited, only about 20% of it is absorbed. Hesperedin, the glycosylated form, on the other hand, is almost completely absorbed [20, 21].”

  1. Wdowiak K, Walkowiak J, Pietrzak R et al. (2022) Bioavailability of Hesperidin and Its Aglycone Hesperetin-Compounds Found in Citrus Fruits as a Parameter Conditioning the Pro-Health Potential (Neuroprotective and Antidiabetic Activity)-Mini-Review. Nutrients 14. https://doi.org/10.3390/nu14132647
  2. Youdim KA, Dobbie MS, Kuhnle G et al. (2003) Interaction between flavonoids and the blood-brain barrier: in vitro studies. J Neurochem 85:180–192. https://doi.org/10.1046/j.1471-4159.2003.01652.x.

“After oral intake, caffeine is 99% absorbed and peaks in the bloodstream after 30-45 minutes [37]. Cafestol is absorbed at about 70%, while kahweol has a slightly higher bioavailability [38]. All three substances show the ability to cross the blood-brain barrier [39].”

  1. Janitschke D, Nelke C, Lauer AA et al. (2019) Effect of Caffeine and Other Methylxanthines on Aβ-Homeostasis in SH-SY5Y Cells. Biomolecules 9. https://doi.org/10.3390/biom9110689
  2. Roos B de, Meyboom S, Kosmeijer-Schuil TG et al. (1998) Absorption and urinary excretion of the coffee diterpenes cafestol and kahweol in healthy ileostomy volunteers. J Intern Med 244:451–460. https://doi.org/10.1111/j.1365-2796.1998.00386.x
  3. Socała K, Szopa A, Serefko A et al. (2020) Neuroprotective Effects of Coffee Bioactive Compounds: A Review. Int J Mol Sci 22. https://doi.org/10.3390/ijms22010107

Unfortunately, we cannot show how exactly the substances are transported from the intestine to the brain

  1. Unpublished data mentioned in the article. If no specific reason, the authors are encouraged to show their unpublished data results in the supplementary materials. These results may also be helpful as a future reference.

Unfortunately, we are not able to publish these data here because they are currently waiting for the review process in another journal.

  1. Combinational effect in a smaller range such as two or three compounds. As the authors mentioned in their article, the combinational effect of two or three compounds may also be necessary to analyze and understand their synergetic effects precisely. The current combinational effect study has included too many compounds and the results shown by the authors cannot clearly show all potential synergetic effects among these compounds. Therefore, the authors cannot sufficiently explain the differences in the cocktail treatment compared to individual compounds. Here I just simply want to emphasize that this point may limit the impact of this article. Hopefully, in the future study, the authors could provide more details.

Indeed, a small amount of substances would better explain the effects. We are aware of this and plan to reduce the number for future studies to achieve the largest possible effect.

  1. Wording issue. The term “SCTR” is not defined in the first place and I found its meaning in the caption for figure 2. If there is no specific reason, it would be better to use straightforward and common words such as treated/cocktail and control to describe your groups. Creating new abbreviations like SC and SCTR could be confusing and lower the readability of your research article. In addition, the authors' first explanation of SCTR is in the caption of their figures, which may not be obvious enough to readers. 

Thank you very much for this reference. Indeed, the definition of the abbreviation SCTR was missing. We have now made up for this in the methods section and thus hope that it is now more understandable.

The overall quality of English is good but multiple typos and grammar errors have been detected. These typos and errors are listed below:

  1. Introduction section, page 1 line 39: “… leading to to reduced…” has repeating words.
  2. Introduction section, page 2 line 47: “The flavonoid is hesperetin (HstP) is an ……” has grammar error.
  3. Introduction section, page 2 line 68: “… towards _-secretase, as well as preventing A_ from crossing ……” has two typos displaying special symbols.
  4. Introduction section, page 2 line 82: “… oxidative stress, A_ production, ……” has a typo displaying a special symbol.
  5. Figure 3, page 5 line 143: “… as the mean +/- SEM. *P>0.05 ……” has a typo. It should be p<0.05.

  1. Discussion section, page 7 line 212-213: “SC was examined for it effects on ……. Although data were did not reach ……” has two grammar errors.

Thank you for the comments, we have corrected the mistakes.

Round 2

Reviewer 1 Report

The Authors made some extensive corrections in text and Figures. The Results are presented clearer and manuscript can be considered for Submission. 

Please, some mistakes on English language and editing are still detectable throughout the text.

Please, check language and editing throughout the text, as lines 24, 53, 211, 265, 274-275.

Author Response

Reviewer 1:

The Authors made some extensive corrections in text and Figures. The Results are presented clearer and manuscript can be considered for Submission. 

Please, some mistakes on English language and editing are still detectable throughout the text.

Please, check language and editing throughout the text, as lines 24, 53, 211, 265, 274-275.

Thank you, for the hints. We corrected the mistakes.
